# A Universal Self-Supervised Paradigm via 3D Gaussian Splatting

## Abstract

Pre-training on large-scale unlabeled datasets has proven effective for enhancing model performance on downstream tasks, particularly when annotated data is scarce. However, due to the inherent discrepancies in data structures across modalities, most existing self-supervised approaches are tailored to either 2D or 3D networks, limiting their generalizability. In this paper, we propose $GS^3$, a 3D Gaussian Splatting (GS)-based universal self-supervised framework, which bridges 2D and 3D modalities and enables pre-training of both 2D and 3D encoders. The core idea is to formulate neural rendering as a pretext task: visual features extracted from input data are used to predict scene-level 3D Gaussians, which are then rendered into images via a fast tile-based rasterizer. The model is optimized by minimizing the difference between rendered and real images, with a masked modeling strategy further encouraging robust and spatially-aware representation learning. We evaluate $GS^3$ across five representative downstream tasks, including detection, segmentation, and reconstruction. Experimental results show that $GS^3$ consistently achieves performance on par with or surpassing state-of-the-art methods, while significantly reducing memory overhead compared to prior NeRF-based approaches.

## 1 Introduction

In recent years, deep neural networks trained with supervised learning have achieved remarkable success across a wide range of vision tasks, such as object detection and segmentation. However, acquiring large-scale, high-quality annotations, especially in the 3D domain, remains expensive and labor-intensive. For example, annotating a single indoor scene with thousands of 3D points can take up to 30 minutes Dai et al. (2017); Armeni et al. (2016). To address this bottleneck, self-supervised learning (SSL) has emerged as a promising paradigm for learning transferable representations from unlabeled data.

Existing SSL approaches can be broadly grouped into three paradigms: completion-based, contrast-based, and rendering-based methods. Completion-based methods He et al. (2022); Zhang et al. (2022); Hess et al. (2023); Liu et al. (2022) learn to reconstruct missing parts of the input, but often suffer from sensitivity to masking strategy and limited semantic understanding. Contrast-based methods Xie et al. (2020); Zhang et al. (2021); Huang et al. (2021) aims to learn invariant features by enforcing consistency across augmented views, yet typically rely on carefully crafted sampling strategies and suffer from slow convergence. Critically, both paradigms are often tailored to a single modality, either 2D or 3D, due to structural discrepancies across data types, thus limiting their generalizability.

Rendering-based SSL offers a potential solution for bridging modalities. Ponder Huang et al. (2023); Zhu et al. (2023) introduces a NeRF-based framework, which back-projects multi-view RGB-D images to construct a 3D feature volume and renders the images via volume rendering for supervision. Although such frameworks have the potential to enable pre-training of both 2D and 3D encoders, they are computationally expensive and face significant challenges when training 2D encoders, as the lack of accurate depth maps introduces ambiguity during back-projection and limits the semantic fidelity of the resulting 3D feature volume.

To address these limitations, we propose **GS**$^3$, an efficient 3D **G**aussian **S**platting-based universal **S**elf-**S**upervised (**GS**$^3$) framework for representation learning. GS$^3$ takes as input two-view observations with overlapping fields of view, and predicts scene-level 3D Gaussians from extracted visual features. These Gaussians are then rendered into 2D images using a fast tile-based rasterizer, avoiding the need for expensive volumetric rendering or explicit feature projection. The model is optimized by minimizing the photometric reconstruction loss between rendered and real images. To further enhance spatial understanding and feature robustness, we introduce a masked modeling strategy, which randomly masks the input modality and encourages the model to reconstruct the complete observation from partial 3D Gaussians. The proposed GS$^3$ framework enables effective pre-training of both 2D

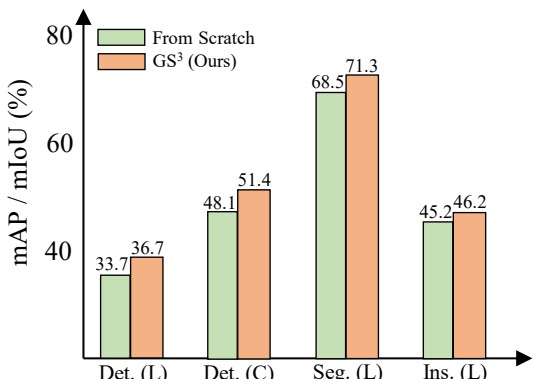

Figure 1: Impact of GS$^3$ pre-training on on four downstream tasks: 3D object detection on SUN RGB-D (L), image-based 3D detection on ScanNet (C), 3D semantic segmentation on S3DIS (L), and 3D instance segmentation on S3DIS (L). Here, L and C denote LiDAR point clouds and camera images, respectively.

image encoders and 3D point cloud encoders, offering a scalable and efficient solution for universal SSL. As shown in Fig. 1, the pre-trained encoders significantly improve performance over training from scratch across multiple downstream tasks. The main contributions are summarized as follows:

- We propose GS$^3$, an efficient 3D Gaussian Splatting-based universal self-supervised learning framework, which can support pre-training of both 2D and 3D encoders.

- GS$^3$ eliminates the need for volumetric rendering or feature projection by directly predicting and rendering 3D Gaussians, resulting in significantly lower memory overhead compared to NeRF-based methods. Moreover, it learns powerful 2D encoders, thanks to its ambiguity-free rendering pipeline.

- Extensive experiments on five representative downstream tasks demonstrate the strong transferability of the learned representations, thus validating the effectiveness of our universal framework.

## 2 RELATED WORK

**Self-supervised learning.** Self-supervised learning (SSL) has emerged as a promising approach for representation learning without human annotations. In 2D vision, SSL approaches are broadly grouped into completion-based and contrast-based methods. Completion-based methods, such as MAE He et al. (2022) and SimMIM Xie et al. (2022), recover masked image patches from remaining patches to learn robust representations. Contrast-based methods Chen et al. (2020); He et al. (2020) learn invariant features by enforcing consistency across augmented views.

In the 3D domain, many SSL methods follow similar paradigms. **Completion-based** approaches Pang et al. (2022); Zhang et al. (2022); Liu et al. (2022) reconstruct masked point clouds or voxel grids from partial observations. For example, PointMAE Pang et al. (2022) reconstructs masked point patches using a transformer-based autoencoder, while VoxelMAE Hess et al. (2023) adapts this paradigm to outdoor LiDAR scenes using voxel representations. MaskPoint Liu et al. (2022) designs a pretext task for binary classification to distinguish between masked and unmasked points. **Contrast-based** methods Xie et al. (2020); Zhang et al. (2021) learn representations invariant to geometric transformations or temporal changes. PointContrast Xie et al. (2020) computes correspondences between two different views of point cloud scene. Other works Huang et al. (2021); Chen et al. (2022) extend SSL to 4D sequences by leveraging spatio-temporal cues. Despite their success, most 2D and 3D SSL methods are modality-specific and cannot be applied across different data types due to data structural discrepancies.

Recently, **rendering-based** frameworks offer a potential solution for bridging 2D and 3D modalities. Ponder Huang et al. (2023); Zhu et al. (2023) back-project RGB-D inputs into 3D space to construct volumetric features, which are then rendered via NeRF-based volume rendering. The model is optimized by minimizing the reconstruction loss between rendered and original images. Although such frameworks have the potential to enable pre-training of both 2D and 3D encoders, they suffer from high memory costs and encounter ambiguous geometry specifically when training 2D encoders. Follow-up works like UniPad Yang et al. (2024) extend this paradigm to outdoor settings. In contrast, our method leverages 3D Gaussian Splatting to achieve lightweight, projection-free rendering, enabling efficient and scalable universal SSL.

**Neural scene representation.** Neural scene representations aims to model 3D geometry and appearance of 3D scenes using neural networks. NeRF Mildenhall et al. (2020) and its extensions Barron et al. (2021); Wang et al. (2021) use MLPs to represent radiance fields and synthesize images via volumetric rendering. However, these methods are computationally intensive due to the large number of ray queries. Recently, 3D Gaussian Splatting (GS) Kerbl et al. (2023) was introduced as a real-time alternative that models scenes using a set of anisotropic Gaussians and renders them via a tile-based rasterizer. Although subsequent works Lee et al. (2024); Yu et al. (2024) have made progress in improving rendering quality and efficiency, they rely on scene-specific optimization. PixelSplat Charatan et al. (2024) initiates generalizable GS-based rendering by predicting 3D Gaussian parameters from multi-view features. MVSplat Chen et al. (2024) constructs a lightweight cost volume to replace the epipolar transformer for cross-view encoding, thereby improving rendering efficiency. FreeSplat Wang et al. (2024) introduces adaptive cost view aggregation and pixel-wise fusion to enable free-view synthesis over a wide range of views. In this work, we explore 3D GS not for image synthesis, but as the backbone for an efficient and universal self-supervised learning framework.

## 3 METHODOLOGY

In this section, we introduce our proposed framework in details. We begin with an overview of the universal framework, followed by detailed discussions for 3D point cloud pre-training and 2D image pre-training. We then introduce a masked modeling strategy, and finally describe our pre-training objectives.

### 3.1 UNIVERSAL FRAMEWORK OVERVIEW

Inspired by the ability of 3D Gaussian Splatting (GS) to bridge the 3D and 2D domains, we propose GS$^3$, a universal self-supervised framework that supports the pre-training of both 2D and 3D encoders. Given a pair of input observations with overlapping fields of view, our framework extracts modality-specific features and predicts a group of scene-level 3D Gaussians. These Gaussians are then rendered into RGB images via a differentiable tile-based rasterizer, thus serving as supervision for encoder pre-training. The overall pipeline is illustrated in Fig. 2 and Fig. 3.

For **3D pre-training**, we first back-project RGB-D input $\{\mathbf{I}_i, \mathbf{D}_i | \mathbf{I}_i \in \mathbb{R}^{H \times W \times 3}, \mathbf{D}_i \in \mathbb{R}^{H \times W}\}_{i=1}^2$ into 3D space to obtain two point clouds $\{\mathbf{P}_i | \mathbf{P}_i \in \mathbb{R}^{(H \times W) \times 3)}\}_{i=1}^2$, followed by a point cloud encoder $f_P$ to extract point-wise features $\mathcal{F}_P = \{\mathcal{F}_P^i\}_{i=1}^2$, $\mathcal{F}_P^i = f_P(\mathbf{P_i})$, where $H$ and $W$ are the height and width of the input image, respectively. For **2D pre-training**, we feed RGB images $\{\mathbf{I}_i\}_{i=1}^2$ into an image encoder $f_I$ to obtain image features $\mathcal{F}_I = \{\mathcal{F}_I^i\}_{i=1}^2$, $\mathcal{F}_I^i = f_I(\mathbf{I}_i)$. To establish cross-view correspondences and integrate complementary geometry and appearance information across views, we adopt an Epipolar Transformer Charatan et al. (2024), which performs cross-attention from one view to another to refine features. This yields encoded features $\hat{\mathcal{F}}_P$ or $\hat{\mathcal{F}}_I$ for 3D or 2D inputs, respectively. Finally, we apply a lightweight feed-forward network $f_\theta$ to transform the encoded features into anisotropic 3D Gaussian parameters (*e.g.*, means $\mu \in \mathbb{R}^3$, covariance matrices $\Sigma \in \mathbb{R}^{3 \times 3}$), thereby producing a compact and expressive 3D scene representation. The predicted Gaussians are rendered into view-dependent images using a differentiable tile-based rasterizer. Specifically, given a camera pose $W$, the 3D Gaussians are projected onto the image plane:

$$\mu_{2D} = \frac{JW\mu_{3D}}{Z}, \quad \Sigma_{2D} = JW\Sigma_{3D}W^\top J^\top, \tag{1}$$

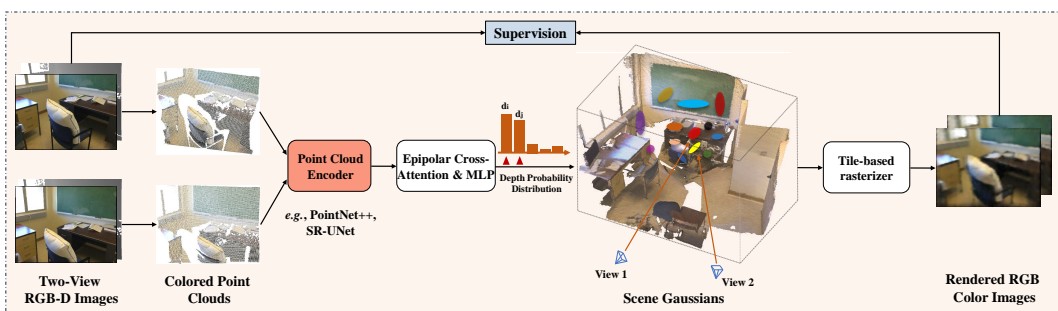

Figure 2: The overall pipeline of GS$^3$ for 3D pre-training with RGB-D inputs. Given a pair of RGB-D images with overlapping views, we back-project them into 3D space to obtain colored point clouds. A point cloud encoder extracts point-wise features, which are then used to predict 3D Gaussians in a point-aligned manner. These Gaussians are rendered into RGB images using a differentiable tile-based rasterizer. The encoder is trained by minimizing the photometric discrepancy between rendered and ground-truth images.

where $Z$ denotes the depth value of the Gaussian and $J$ is the Jacobian matrix for the perspective projection. For each pixel $v$, we compute the alpha value of the $i$-th Gaussian as $\alpha_i = o_i G_i^{2D}(v)$, where $G_i^{2D}$ is the 2D Gaussian projected onto the image plane. The final pixel color is computed as:

$$C(v) = \sum_{i \in \mathcal{N}} \mathbf{c}_i \alpha_i \prod_{j=1}^{i-1} (1 - \alpha_j). \tag{2}$$

Both the 2D and 3D encoders are pre-trained by minimizing the photometric loss between the rendered images and ground-truth images, which encourages the encoders to learn robust and discriminative representations.

## 3.2 PRE-TRAINING FOR 3D POINT CLOUDS

Our 3D pre-training pipeline takes as input a pair of RGB-D images with known camera intrinsics and extrinsics. As shown in Fig. 2, we back-project the RGB-D images into 3D space to construct point clouds, which serve as input to a 3D encoder. We adopt both a point-based architecture (*e.g.*, PointNet++ Qi et al. (2017)) and a discretization-based encoder (*e.g.*, SR-UNet Xie et al. (2020)) to validate the generality of our framework.

Each 3D point corresponds to a pixel in the input RGB-D image, allowing us to establish a dense, point-aligned representation. Given the encoded point-wise features $\hat{\mathcal{F}}_{\mathcal{P}}$, we define a mapping function $f_\theta$ to predict the parameters of $k$ 3D Gaussians per point, including the depth logits $d$, offset values $\delta$, covariance matrices $\Sigma$, alpha values $\alpha$ and spherical harmonics coefficients $\mathbf{c}$. Since the viewing ray direction of each pixel is known, estimating the depth is sufficient to recover the 3D location of each predicted Gaussian.

$$f_\theta : \left\{ \hat{\mathcal{F}}_i \right\}_{i=1}^2 \mapsto k \times \left\{ (d_j, \delta_j, \mathbf{\Sigma}_j, \alpha_j, \mathbf{c}_j) \right\}_{j=1}^{(H \times W) \times 2}. \tag{3}$$

To infer the 3D locations of Gaussians, we uniformly discretize the depth range into $Z$ bins and predict a categorical distribution over the candidates. The final depth is obtained by selecting the top-$k$ depth bins with highest probability and refining them with offset values $\delta_j \in [0, 1]^Z$. This formulation enables point-aligned Gaussian prediction while maintaining geometric consistency across views. Given two input views of size $H \times W$, we generate a total of $k \times (H \times W) \times 2$ Gaussians per sample. Unlike traditional 3D GS methods that require scene-specific optimization, our feed-forward formulation supports batched training across multiple scenes, making it scalable and compatible with self-supervised learning.

In addition to depth-based generation, we also incorporate direct positional cues from the 3D point clouds. Specifically, we reformulate the Gaussian center prediction using positional offsets:

$$f_\theta : \left\{ \hat{\mathcal{F}}_P^i \right\}_{i=1}^2 \mapsto k \times \left\{ (\Delta\mu_j, \mathbf{\Sigma}_j, \alpha_j, \mathbf{c}_j) \right\}_{j=1}^{(H \times W) \times 2}, \tag{4}$$

$$\mu_j = p_i + \Delta\mu_j, \tag{5}$$

where $p_i$ denotes the original 3D coordinate of the $i$-th point, and $\Delta\mu_j$ is the learned offset to its associated Gaussian center $\mu_j$. This alternative formulation further leverages the rich geometric information embedded in 3D point clouds obtained by RGB-D back-projection.

## 3.3 PRE-TRAINING FOR 2D IMAGES

To fully unify 2D and 3D modalities within a single self-supervised framework, our GS[3] supports the pre-training of both image and point cloud encoders under the same rendering-based objective. The pipeline for 2D image pre-training follows the same formulation as the 3D counterpart, ensuring structural symmetry and consistent supervision.

As shown in Fig. 3, we take two-view RGB images as input and employ ResNet-50 He et al. (2016) as the 2D encoder to extract pixel-level features. These features are then enhanced via an Epipolar Transformer for cross-view encoding. Following the same depth-based Gaussian generation strategy as in the 3D pre-training, we use a feed-forward network to predict per-pixel depth logits and offset values, which determine the 3D centers of the Gaussians along the viewing rays. Combined with additional parameters such as covariance, opacity, and spherical harmonics coefficients, this yields a dense,

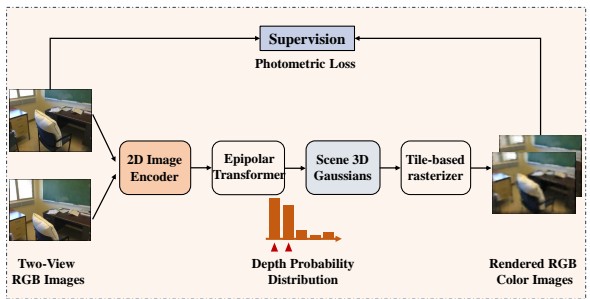

Figure 3: The overall pipeline of GS[3] for 2D pre-training with RGB image inputs. This pipeline shares the same rendering formulation as its 3D counterpart, enabling unified and modality-agnostic representation learning.

pixel-aligned set of 3D Gaussians from RGB-only inputs. These Gaussians are subsequently rendered into RGB images via a differentiable tile-based rasterizer.

Although the architectural pipeline is structurally aligned with that of point cloud pre-training, the 2D pathway reinforces the modality-unified design and ensures transferability to image-based tasks. Together, the 2D and 3D pipelines constitute a cohesive and modality-agnostic framework for universal representation learning.

## 3.4 MASKED MODELING STRATEGY

As discussed above, our method predicts scene-level 3D Gaussians directly from visual features, with each pixel or 3D point contributing one or more Gaussian primitives. However, this dense one-to-one (or one-to-many) mapping can lead to redundant Gaussians, as similar local structures often generate overlapping primitives. Such redundancy increases computational overhead and may impair representation learning by overfitting to repetitive geometric patterns.

To alleviate this issue, we introduce a masked modeling strategy tailored for our rendering-based framework. Inspired by completion-based approaches Liu et al. (2022), we randomly mask 50% of the input points or pixels. However, instead of reconstructing the masked regions, we use only the visible subset to predict scene Gaussians, which are then rendered into RGB images as supervision. This design enforces the encoder to infer missing structures from partial observations, encouraging it to capture more holistic and distinctive features. Compared to naive dense training, our masked strategy not only improves feature robustness but also reduces Gaussian redundancy by promoting compact and informative representations.

## 3.5 PRE-TRAINING OBJECTIVES

To ensure consistent supervision across modalities, our self-supervised framework employs a unified set of pre-training objectives for both 2D and 3D encoders. Specifically, we avoid using depth maps as explicit supervision and instead rely solely on photometric reconstruction losses, enabling modality-agnostic pre-training within a unified framework. Our total pre-training loss $L$ is

a weighted combination of an image reconstruction loss $L_{\text{color}}$ and a LPIPS Zhang et al. (2018) loss $L_{\text{lpips}}$, *i.e.*, $L = L_{\text{color}} + \lambda \cdot L_{\text{lpips}}$.

**Image reconstruction Loss** $L_{color}$: This loss measures pixel-wise color consistency between the rendered image and ground-truth images. We adopt the mean squared error (MSE) over all pixels:

$$L_{\text{color}} = \frac{1}{H \times W} \sum_{i=1}^{H \times W} (I_r(i) - I_{\text{gt}}(i))^2, \qquad (6)$$

where $I_r$ and $I_{\text{gt}}$ denote the rendered and ground-truth RGB images, respectively, and $H \times W$ is the number of pixels.

**LPIPS loss** $L_{\textbf{lpips}}$: To capture high-level perceptual similarity, we incorporate the LPIPS loss Zhang et al. (2018), which compares feature maps extracted from a pre-trained VGG network Simonyan & Zisserman (2014):

$$L_{\text{lpips}} = \sum_l \frac{1}{M_l} \sum_{i=1}^{M_l} \|w_l \odot (\hat{f}_l(I_r(i)) - \hat{f}_l(I_{gt}(i)))\|_2^2, \qquad (7)$$

where $\hat{f}_l(\cdot)$ is the normalized feature map from the $l$-th layer, $w_l$ denotes the channel-wise weighting, and $M_l = H_l \times W_l$ is the number of spatial locations in the feature map at layer $l$. In our work, $\lambda$ is set to 0.05.

## 4 EXPERIMENTS

### 4.1 EXPERIMENTAL SETTINGS

**Datasets.** We conduct all pre-training experiments on the ScanNet v2 dataset Dai et al. (2017), which contains a total of 1513 indoor scenes captured in diverse real-world environments. Following standard protocol, we use 1201 scenes for training and reserve 312 scenes for testing. Each scene comprises hundreds of temporally continuous RGB-D images along with the corresponding camera intrinsic parameters and poses. Rich 3D annotations, including point-level segmentation, instance-level segmentation, and 3D bounding boxes, are provided but not used during pre-training.

**Implementation details.** Our self-supervised framework accepts two-view RGB-D images with overlapping fields of view. For 3D pre-training, we back-project the RGB-D inputs into 3D space to obtain point clouds. For 2D pre-training, we use the RGB images directly. All input images are resized to a resolution of $320 \times 240$, with a fixed frame interval of 5 between views. We evaluate two types of point cloud encoders: the point-based PointNet++ and the voxel-based SR-UNet, both configured to output 128-dimensional features. For the image pre-training, we adopt ResNet-50 as the 2D encoder. All encoders are trained from scratch during pre-training. Additional architectural details are provided in the supplementary material.

We pre-train our model for 100 epochs with a batch size of 4, where each batch corresponds to a single scene. All experiments are conducted on a single NVIDIA A100 GPU with 40GB memory. We adopt the AdamW optimizer Loshchilov (2017) with an initial learning rate of 1e-4 and a weight decay of 0.05. A cosine annealing schedule Loshchilov & Hutter (2016) is used to progressively decay the learning rate to a minimum of 1e-6. To obtain diverse training samples, we apply the same random rotations to both the point clouds and camera poses along the X, Y, and Z axes. The rotation angles are uniformly sampled from $[-\pi/64, \pi/64]$ for the X and Y axes, and from $[-\pi, \pi]$ for the Z axis. For downstream tasks, we initialize the encoder with the pre-trained weights and follow the original protocols for fine-tuning. In addition, **we adopt baseline methods widely used in recent works to ensure fair and reproducible comparisons.**

### 4.2 FINE-TUNING ON DOWNSTREAM TASKS

To evaluate the effectiveness and generalizability of our proposed GS$^3$ framework, we pre-train the feature encoders on the ScanNet v2 dataset and transfer the learned weights for downstream tasks.

**3D object detection.** We use the ScanNet v2 and SUN RGB-D dataset Song et al. (2015) to evaluate the transferability of our pre-trained point cloud encoder on the 3D object detection task. This

Table 1: Comparative 3D object detection results among current self-supervised methods.

| Method | Detector | mAP@0.5 | mAP@0.25 |
|---|---|---|---|
| SUN RGB-D | | | |
| VoteNet | - | 33.7 | 57.7 |
| STRL | VoteNet | 35.0 | 58.2 |
| RandomRooms | VoteNet | 35.4 | 59.2 |
| PointContrast | VoteNet | 34.8 | 57.5 |
| PC-FractalDB | VoteNet | 33.9 | 59.4 |
| DepthContrast | VoteNet | 35.4 | 60.4 |
| IAE | VoteNet | 36.0 | 60.4 |
| Ponder-RGBD | VoteNet | 36.6 | 61.0 |
| **GS$^3$-RGBD** | VoteNet | **36.7 (+3.0)** | **61.3 (+3.6)** |
| ScanNet v2 | | | |
| FCAF3D | - | 57.3 | 71.5 |
| **GS$^3$-RGBD** | FCAF3D | **59.4 (+2.1)** | **73.1 (+1.6)** |

Table 2: Comparison of pre-training memory cost.

| Method | #Rendering Pixels | Pre-training Memory (GB/batch) ↓ |
|---|---|---|
| Ponder-RGBD | 4,800 | 38.4 |
| **GS$^3$-RGBD** | 320×240 | **10.3** |

Table 3: Comparative **Image-based 3D object detection** results on the ScanNet v2 dataset.

| Method | mAP@0.25 |
|---|---|
| ImVoxelNet | 48.1 |
| NeRF-based pre-training | 50.2 |
| **GS$^3$-RGB** | **51.4 (+3.3)** |

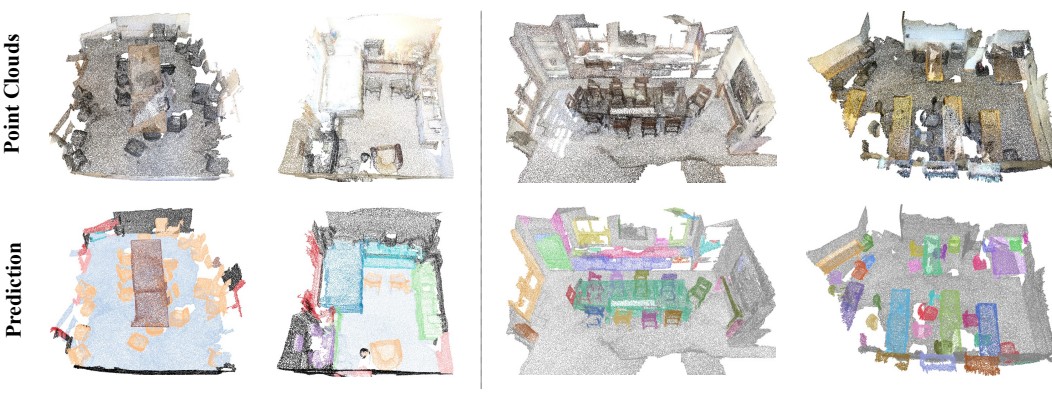

(a) 3D semantic segmentation   (b) 3D instance segmentation

Figure 4: Qualitative results of our fine-tuned model on downstream (a) 3D semantic segmentation and (b) 3D instance segmentation tasks.

dataset contains 10,335 indoor scenes, each of which provides RGB-D images, camera poses and 3D box annotations. Following prior works, we adopt VoteNet Qi et al. (2019) and FCAF3D Rukhovich & et al. (2022) as the baseline detectors and evaluate using mean Average Precision (mAP) at IoU thresholds of 0.25 and 0.5. Table 1 presents a comprehensive comparison with existing 3D self-supervised learning methods. Our method, GS$^3$-RGBD, significantly boosts the performance of VoteNet, achieving remarkable gains of +3.0% and +3.6% in mAP@0.5 and mAP@0.25, respectively. Compared to the NeRF-based Ponder-RGBD Huang et al. (2023), GS$^3$-RGBD achieves slightly higher performance (+0.1% at mAP@0.5 and +0.3% at mAP@0.25). Consistent improvements are also observed when using a more advanced baseline detector, FCAF3D Rukhovich & et al. (2022). In addition to performance, we compare the pre-training memory cost in Table 2. Our GS$^3$ only consumes 10.3 GB per batch (at a resolution of 320×240), which is approximately **3.7×** less than the 38.4 GB required by Ponder-RGBD with 4,800 rendering pixels. These results demonstrate that Our GS$^3$-RGBD not only learns highly transferable representations for downstream tasks, but also offers a significantly more memory-efficient training pipeline. Thus, our framework serves as an effective and lightweight alternative to NeRF-based method.

**Image-based 3D object detection.** To assess the transferability of our pre-trained image encoder, we evaluate it on the image-based 3D object detection task using the ScanNet v2 dataset. ImVoxelNet Rukhovich et al. (2022) is adopted as the baseline detector. For a fair comparison, we also implement a NeRF-based rendering framework to pre-train the image encoder. As shown in Table 3, incorporating GS$^3$ into ImVoxelNet yields a notable improvement of 3.3% in mAP@0.25 over the baseline. In addition, GS$^3$ surpasses the NeRF-based approach by 1.2%. NeRF-based frameworks struggle to pre-train 2D encoders effectively due to the absence of depth maps, which hinders accurate back-projection and limits the Discriminability of the learned representations. In contrast,

Table 4: **3D semantic segmentation** results on the S3DIS and ScanNet v2 datasets. † denotes the reproduced results.

| Method | S3DIS (Area-5) | | ScanNet v2 | |
|---|---|---|---|---|
| | mIoU↑ | mAcc↑ | mIoU↑ | mAcc↑ |
| PointNet | 41.1 | 49.0 | - | - |
| PointNet++ | - | - | 53.5 | - |
| KPConv | 67.1 | 72.8 | 69.2 | - |
| MSC | - | - | 75.3 | - |
| MinkUNet | - | - | 72.2 | - |
| Ponder + MinkUNet | - | - | 73.5 | - |
| MinkUNet† (5cm) | 62.8 | 70.6 | 66.6 | 75.0 |
| + GS³-RGBD | **64.9** (+2.1) | **71.8** (+1.2) | **68.5** (+1.9) | **76.9** (+1.9) |
| MinkUNet† (2cm) | 68.5 | 75.2 | 71.9 | 80.6 |
| + GS³-RGBD | **71.3** (+2.8) | **76.7** (+1.5) | **74.0** (+2.1) | **81.6** (+1.0) |

Table 5: **3D instance segmentation** results on the S3DIS and ScanNet v2 dataset. † denotes the reproduced results.

| Method | S3DIS (Area-5) | | ScanNet v2 | |
|---|---|---|---|---|
| | avg. AP | AP@0.5 | avg. AP | AP@0.5 |
| 3D-SIS | - | - | - | 18.7 |
| GSPN | - | - | 19.3 | 37.8 |
| PointGroup | - | 57.8 | 34.8 | 56.7 |
| DyCo3D | - | - | 35.4 | 57.6 |
| CSC | - | - | - | 59.4 |
| MSC | - | - | - | 59.5 |
| PointGroup† (5cm) | 40.1 | 55.7 | 27.2 | 49.1 |
| +GS³-RGBD | **40.8** (+0.7) | **57.9** (+2.2) | **28.5** (+1.3) | **51.0** (+1.9) |
| PointGroup† (2cm) | 45.2 | 59.4 | 35.2 | 57.6 |
| +GS³-RGBD | **46.2** (+1.0) | **61.9** (+2.5) | **37.4** (+2.2) | **59.5** (+1.9) |

Table 6: Image-based 3D scene reconstruction results on the ScanNet v2 dataset. † denotes the reproduced results.

| Method | Acc | Comp | F-Score |
|---|---|---|---|
| Atlas† | 0.155 | 0.114 | 0.497 |
| NeRF-based pre-training | 0.147 | 0.109 | 0.509 |
| **GS³-RGB (Ours)** | **0.139** | **0.104** | **0.535** |

Table 7: Ablation study on the number of input views. 3D semantic segmentation mIoU and mAcc on S3DIS.

| #View | mIoU | mAcc |
|---|---|---|
| MinkUNet | 68.5 | 75.2 |
| 2 | 71.3 (**+2.8**) | 76.7 (**+1.5**) |
| 3 | 71.8 (**+3.3**) | 77.0 (**+1.8**) |

our GS³ framework explicitly models 3D Gaussians with depth-aware generation, enabling more precise and transferable representation learning from RGB-only inputs.

**3D semantic segmentation.** We evaluate the semantic segmentation performance of our fine-tuned model on two widely-used indoor benchmarks: ScanNet v2 and S3DIS Armeni et al. (2016). We adopt MinkUNet Choy et al. (2019) as baseline, and report mean Intersection-over-Union (mIoU) and mean Accuracy (mAcc) as the primary evaluation metrics. Table 4 presents the quantitative results of GS³ pre-training combined with MinkUNet and PT. Our method consistently boosts the performance across both datasets and voxel resolutions. Specifically, with a voxel size of 2cm, MinkUNet with GS³ improves the mIoU by +2.8% on S3DIS and +2.1% on ScanNet v2. Similar gains are observed in mAcc, with improvements of +1.5% and +1.0% respectively. Notably, our method also slightly outperforms the NeRF-based framework Ponder Huang et al. (2023) on ScanNet v2 (74.0% vs. 73.5% in mIoU), while requiring significantly less memory during pre-training. These results confirm the effectiveness and generalizability of our pre-trained representations, which transfer well to semantic segmentation tasks across different data modalities and scene capture protocols. Qualitative visualization results on ScanNet v2 are provided in Fig. 4(a).

**3D instance segmentation.** We evaluate the transferability of our GS³ framework to the task of 3D instance segmentation on the S3DIS and ScanNet v2 datasets. We adopt the classic PointGroup Jiang et al. (2020) as the baseline model, and follow standard evaluation protocols using average precision (AP) and AP at an IoU threshold of 0.5 (AP@0.5). The average AP is computed by averaging the AP values across IoU thresholds ranging from 50% to 95% with a 5% interval. Table 5 presents the quantitative results. Our GS³ framework consistently improves the performance of PointGroup across different voxel resolutions. Specifically, with a voxel size of 2cm, GS³ attains gains of +1.0% in average AP and +2.5% in AP@0.5 on the S3DIS dataset. Similar improvements are also observed on the ScanNet v2 dataset, demonstrating the effectiveness and generalizability of our pre-trained representations. Qualitative results on ScanNet v2 are visualized in Fig. 4(b).

**Image-based scene reconstruction.** We evaluate the effectiveness of our pre-trained image encoder on the image-based 3D scene reconstruction task using the ScanNet v2 dataset. We adopt the classic Atlas framework Murez et al. (2020) as the baseline. As shown in Table 6, our GS³ significantly outperforms both the baseline and the NeRF-based framework, demonstrating the superiority of GS³ in learning transferable representations for 2D image encoders.

Table 8: Ablation study on mask ratio. 3D semantic segmentation mIoU and mAcc on S3DIS *Area-5*.

| Mask ratio | mIoU | mAcc |
|---|---|---|
| MinkUNet | 68.5 | 75.2 |
| 0% | 70.2 (**+1.7**) | 75.7 (**+0.5**) |
| 25% | 70.7 (**+2.2**) | 76.1 (**+0.9**) |
| **50%** | **71.3 (+2.8)** | **76.7 (+1.5)** |
| 75% | 70.6 (**+2.1**) | 76.2 (**+1.0**) |
| 90% | 69.5 (**+1.0**) | 75.3 (**+0.1**) |

Table 9: Ablation study on input image resolution. 3D semantic segmentation mIoU and mAcc on S3DIS *Area-5*. Input image resolution is in the form of width $\times$ height.

| Resolution | mIoU | mAcc |
|---|---|---|
| MinkUNet | 68.5 | 75.2 |
| $256 \times 192$ | 70.4 (**+1.9**) | 76.1 (**+0.9**) |
| $320 \times 240$ | 71.3 (**+2.8**) | 76.7 (**+1.5**) |
| $512 \times 384$ | 71.5 (**+3.0**) | 76.6 (**+1.4**) |

### 4.3 ABLATION STUDY

We conduct a group of ablation experiments to validate the design choices and hyperparameter settings of our GS$^3$ framework. All experiments are performed on the 3D semantic segmentation task using the S3DIS *Area-5* set.

**Number of input views.** Our GS$^3$ framework adopts two-view image observations with overlapping fields of view during pre-training to generate 3D Gaussians for image rendering. In this ablation, we examine the impact of varying the number of input views on downstream segmentation performance. Table 7 reports the results with different number of input views. We find that the best performance is achieved with three input views, likely due to improved rendering quality and stronger supervision signals from multi-view consistency. However, increasing the number of views substantially raises the computational and memory cost during pre-training. Therefore, we adopt a two-view setting in all experiments to strike a balance between efficiency and performance.

**Mask ratio.** To enhance the representation learning of the point cloud encoder, we introduce a masked modeling strategy that randomly masks portions of the input and encourages the model to learn contextual information. We study the effect of different mask ratios, ranging from 0% to 90%, and report the results in Table 8. Our method achieves the best results with a mask ratio of 50%, producing 71.3% mIoU and 76.7% mAcc. We suggest that overly high masking rates retain too few visible Gaussians, hindering effective geometry and appearance modeling, while too low masking leads to redundancy and limits regularization. Overall, our GS$^3$ consistently improves over the baseline across different mask ratios, indicating that our framework is robust and not sensitive to the masking rate.

**Input image resolution.** In GS$^3$, the input RGB images are back-projected into 3D space to facilitate self-supervised training of the point cloud encoder. A higher input resolution provides finer-grained 2D supervision and denser 3D geometry, potentially benefiting downstream performance. To evaluate this effect, we conduct an ablation study with varying image resolutions. As shown in Table 9, increasing the resolution consistently improves segmentation results, confirming the importance of high-fidelity supervision. However, this comes at the cost of increased memory and computational overhead during pre-training. To balance performance and efficiency, we adopt a resolution of $320 \times 240$ in our experiments.

## 5 CONCLUSION

In this paper, we propose GS$^3$, an efficient 3D Gaussian Splatting (GS)-based universal self-supervised framework for representation learning. By leveraging efficient 3D Gaussian-based rendering, GS$^3$ enables the pre-training of both 2D and 3D encoder within a unified framework, effectively bridging the gap between image and point cloud modalities. Compared to NeRF-based frameworks, GS$^3$ eliminates the need for volumetric rendering and explicit feature projection, resulting in significantly reduced memory overhead and more effective representation learning. Extensive experiments on five downstream tasks demonstrate the strong transferability of the learned representations. We hope GS$^3$ will inspire further research on efficient and universal self-supervised learning across modalities.

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

## A APPENDIX

### A.1 USE OF LARGE LANGUAGE MODELS

We simply use Large Language Models (LLMs) to polish our writing of our full manuscript.

