# A Universal Self-Supervised Paradigm via 3D Gaussian Splatting

## 1 More Experimental Results

In this section, we provide detailed per-category results for downstream tasks to further validate the effectiveness of our GS$^3$ framework.

**Fine-tuning on 3D scene reconstruction.** We evaluate the effectiveness of our fine-tuned model on the 3D scene reconstruction task using the Synthetic Indoor Scene dataset Peng et al. (2020). Following the protocol in Huang et al. (2023), we adopt the widely used ConvONet Peng et al. (2020) as our baseline. The evaluation is conducted using three standard metrics: volumetric IoU, normal consistency (NC), and F-score with a threshold of 1%.

Table 1 reports the quantitative results of our GS$^3$ framework, along with the baseline and several self-supervised methods. Our method achieves competitive performance, with a volumetric IoU of 80.4% and a F-score of 92.6, outperforming the baseline ConvONet by 2.6% and 2.0% respectively. These results demonstrate the effectiveness and transferability of the learned representations from GS$^3$ in enhancing downstream reconstruction performance. In addition, compared to other self-supervised approaches, our 3D Gaussian Splatting-based framework surpasses the contrast-based method IAE Yan et al. (2023) by a significant margin of 4.7% in volumetric IoU, while achieving slightly better performance than the NeRF-based framework Ponder Huang et al. (2023); Zhu et al. (2023). These results highlights the strong potential of our GS$^3$ framework in representation learning.

| Method | Encoder | IoU↑ | NC↑ | F-Score↑ |
|---|---|---|---|---|
| ConvONet Peng et al. (2020) | PointNet++ | 77.8 | 88.7 | 90.6 |
| IAE Yan et al. (2023) | PointNet++ | 75.7 | 88.7 | 91.0 |
| Ponder-RGBD Huang et al. (2023) | PointNet++ | 80.2 | 89.3 | 92.0 |
| **GS$^3$-RGBD** | PointNet++ | 80.4 **(+2.6)** | 89.5 **(+0.8)** | 92.6 **(+2.0)** |

Table 1: Comparative **3D scene reconstruction** results on the Synthetic Indoor Scene dataset.

**Per-category 3D object detection.** Table 2 reports the average precision (AP) for each object category on the SUN RGB-D dataset. Our GS$^3$ framework significantly enhances the overall performance of the baseline VoteNet, yielding a 3.0% gain in mAP@0.5. Notably, GS$^3$ improves the detection accuracy in 8 out of 10 categories, demonstrating consistent benefits across diverse object classes.

**Per-category 3D semantic segmentation.** Table 3 and Table 4 present the mean IoU (mIoU) scores for each semantic class on the S3DIS and ScanNet v2 datasets, respectively. With a voxel resolution of 2cm, MinkUNet pre-trained by our GS$^3$ achieves gains in 9 out of 13 classes on S3DIS (Area-5) and 18 out of 20 classes on ScanNet v2. Similar trends are observed when the voxel size is increased to 5cm, further highlighting the effectiveness of our framework.

**Per-category 3D instance segmentation.** Table 5 and Table 6 show the AP@0.5 results for each instance category on the S3DIS and ScanNet v2 datasets, respectively. Our GS$^3$ framework consistently improves the instance segmentation performance across most categories on both datasets, demonstrating its effectiveness in learning transferable representations.

| Method | mAP@0.5 | bathtub | bed | bookshelf | chair | desk | dresser | nightstand | sofa | table | toilet |
|---|---|---|---|---|---|---|---|---|---|---|---|
| VoteNet Qi et al. (2019) | 33.7 | 47.0 | 50.1 | 7.2 | 53.9 | 5.3 | 11.5 | 40.7 | 42.4 | 19.5 | 59.8 |
| **GS³ + VoteNet** | 36.7 | 54.7 | 53.0 | 10.0 | 53.9 | 7.5 | 17.8 | 40.3 | 51.1 | 17.6 | 61.1 |
| | (+3.0) | (+7.7) | (+2.9) | (+2.8) | (+0.0) | (+2.2) | (+6.3) | (-0.4) | (+8.7) | (-1.9) | (+1.3) |

Table 2: Comparative 3D object detection results for each category on the SUN-RGBD dataset, evaluated with mAP@0.5. The number in each bracket denotes the performance improvement (shown in red) or degradation (shown in blue) compared to the corresponding baseline.

| Method | mIoU | mAcc | ceil. | floor | wall | beam | col. | wind. | door | chair | table | book. | sofa | board | clut. |
|---|---|---|---|---|---|---|---|---|---|---|---|---|---|---|---|
| PointNet Qi et al. (2017) | 41.1 | 49.0 | 88.8 | 97.3 | 69.8 | 0.1 | 3.9 | 46.3 | 10.8 | 52.6 | 58.9 | 40.3 | 5.9 | 26.4 | 33.2 |
| KPConv Thomas et al. (2019) | 67.1 | 72.8 | 92.8 | 97.3 | 82.4 | 0.0 | 23.9 | 58.0 | 69.0 | 91.0 | 81.5 | 75.3 | 75.4 | 66.7 | 58.9 |
| MinkUNet (5cm) Choy et al. (2019) | 65.4 | 71.7 | 91.8 | 98.7 | 86.2 | 0.0 | 34.1 | 48.9 | 62.4 | 89.8 | 81.6 | 74.9 | 47.2 | 74.4 | 58.6 |
| Point Transformer Zhao et al. (2021) | 70.4 | 76.5 | 94.0 | 98.5 | 86.3 | 0.0 | 38.0 | 63.4 | 74.3 | 82.4 | 89.1 | 80.2 | 74.3 | 76.0 | 59.3 |
| MinkUNet† (5cm) Choy et al. (2019) | 62.8 | 70.6 | 90.8 | 96.1 | 81.4 | 0.1 | 18.8 | 53.3 | 60.7 | 84.9 | 75.8 | 69.1 | 61.8 | 68.5 | 55.0 |
| **GS³ + MinkUNet (5cm)** | 64.9 | 71.8 | 91.8 | 99.4 | 83.5 | 0.1 | 26.3 | 53.1 | 54.7 | 88.7 | 76.5 | 71.1 | 65.6 | 76.1 | 56.7 |
| | (+2.1) | (+1.2) | (+1.0) | (+3.3) | (+2.1) | (+0.0) | (+7.5) | (-0.2) | (-6.0) | (+3.8) | (+0.7) | (+2.0) | (+3.8) | (+7.6) | (+1.7) |
| MinkUNet† (2cm) Choy et al. (2019) | 68.5 | 75.2 | 91.6 | 97.6 | 84.1 | 0.0 | 24.5 | 60.3 | 77.5 | 87.8 | 81.6 | 72.6 | 73.8 | 80.3 | 59.0 |
| **GS³ + MinkUNet (2cm)** | 71.3 | 76.7 | 93.3 | 99.9 | 86.5 | 0.1 | 36.0 | 62.9 | 77.8 | 92.3 | 83.3 | 74.1 | 75.3 | 83.0 | 61.7 |
| | (+2.8) | (+1.5) | (+2.3) | (+2.3) | (+2.4) | (+0.1) | (+11.5) | (+2.6) | (+0.3) | (+4.5) | (+1.7) | (+1.5) | (+1.5) | (+2.7) | (+2.7) |

Table 3: Comparative 3D semantic segmentation results for each category on the S3DIS (Area-5) dataset. † denotes the reproduced results. The number in each bracket denotes the performance improvement (shown in red) or degradation (shown in blue) compared to the corresponding baseline.

## 2 VISUALIZATION OF THE SR-UNET AND POINTNET++ ENCODER

The proposed GS³ framework is designed to accommodate a variety of point cloud encoders, including both point-based and discretization-based architectures. In our implementation, we adopt PointNet++ (point-based) and SR-UNet (discretization-based) as the backbone encoders for both the pre-training and fine-tuning stages.

We first present the architecture of SR-UNet, as shown in Fig. 1(a). SR-UNet follows the classical U-Net encoder–decoder design and is primarily constructed using Sparse Convolution (SpConv) and Sparse Deconvolution (SpDeconv) layers. The encoder network consists of five SpConv blocks, while the decoder contains four SpDeconv blocks. Each block adopts the standard 2D ResNet-style design, where each (de)convolution layer is followed by a Batch Normalization (BN) layer and a ReLU activation function.

The architecture of PointNet++ is shown in Fig. 1(b). It consists of four Set Abstraction (SA) layers followed by four Feature Propagation (FP) layers. The number of sampled points and the corresponding radii for the SA layers are set to $[2048, 1024, 512, 256]$ and $[0.2, 0.4, 0.8, 1.2]$, respectively.

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

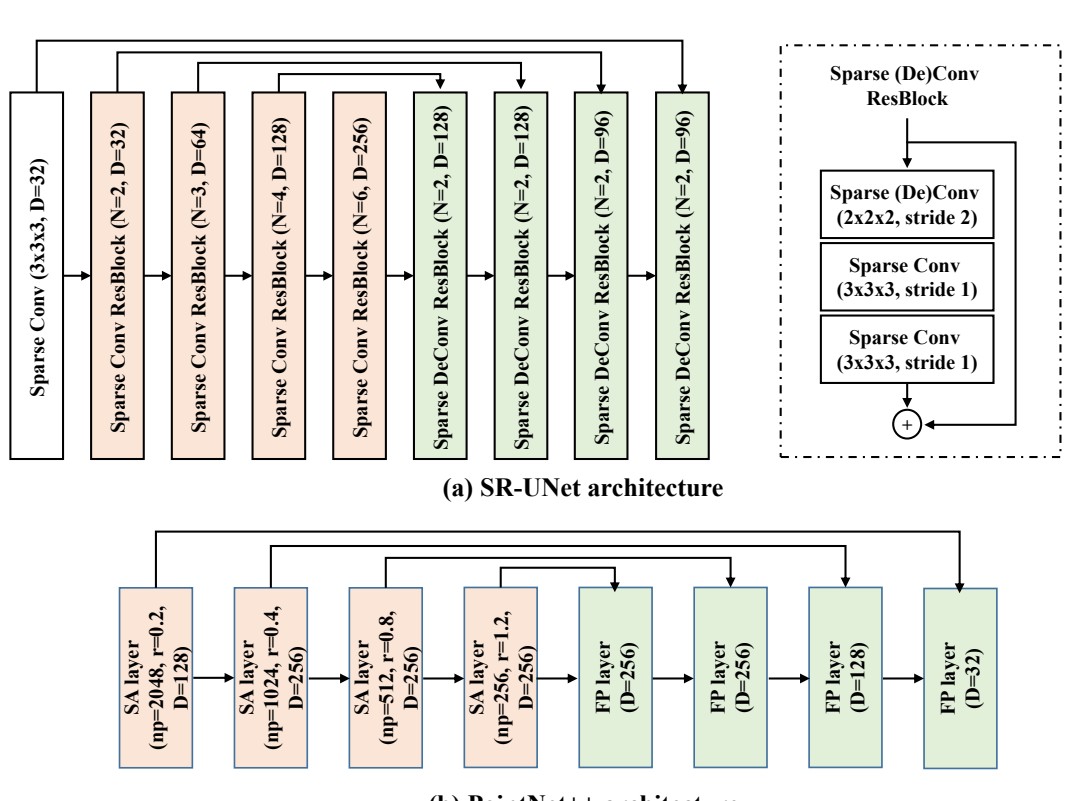

Figure 1: The network architectures of the feature encoders used in our framework. (a) SR-UNet and (b) PointNet++. For SR-UNet, each sparse (de)convolution layer is followed by a BatchNorm (BN) layer and a ReLU activation. $D$ denotes the output feature dimension and $N$ represents the number of repeated blocks. For PointNet++, SA refers to the set abstraction layer and FP to the feature propagation layer. $np$ and $r$ represent the number of sampled points and the radius used in each SA layer, respectively.

| Method | mIoU | mAcc | wall | floor | cabinet | bed | chair | sofa | table | door | window | bookshelf | picture | counter | desk | curtain | refrigerator | shower curtain | toilet | sink | bathtub | other furniture |
|---|---|---|---|---|---|---|---|---|---|---|---|---|---|---|---|---|---|---|---|---|---|---|
| MinkUNet† (2cm) Choy et al. (2019) | 71.9 | 80.6 | 85.8 | 96.3 | 65.7 | 79.5 | 89.9 | 84.5 | 71.3 | 65.4 | 60.3 | 79.4 | 35.3 | 64.9 | 63.0 | 73.0 | 54.5 | 68.0 | 93.1 | 66.3 | 85.2 | 57.0 |
| GS³ + MinkUNet (2cm) | 74.0 (+2.1) | 81.6 (+1.0) | 86.2 (+0.4) | 99.1 (+2.8) | 68.6 (+2.9) | 82.8 (+3.3) | 91.0 (+1.1) | 86.1 (+1.6) | 74.6 (+3.3) | 68.7 (+3.3) | 62.4 (+2.1) | 84.1 (+4.7) | 29.4 (-5.9) | 66.4 (-1.5) | 69.6 (+6.6) | 76.0 (+3.0) | 57.2 (+2.7) | 68.8 (+0.8) | 93.2 (+0.1) | 67.6 (+1.3) | 87.9 (+2.7) | 60.2 (+3.2) |

Table 4: Comparative 3D semantic segmentation results for each category on the ScanNet v2 val set. † denotes the reproduced results. The number in each bracket denotes the performance improvement (shown in red) or degradation (shown in blue) compared to the corresponding baseline.

| Method | AP@50 | ceil. | floor | wall | beam | col. | wind. | door | chair | table | book. | sofa | board |
|---|---|---|---|---|---|---|---|---|---|---|---|---|---|
| PointGroup† (5cm) Jiang et al. (2020) | 55.7 | 46.2 | 95.5 | 64.0 | 0.0 | 37.1 | 72.1 | 55.0 | 64.3 | 29.6 | 35.4 | 88.4 | 80.6 |
| GS³ + PointGroup (5cm) | 57.9 (+2.2) | 45.3 (-0.9) | 98.8 (+3.3) | 65.5 (+1.5) | 0.0 (+0.0) | 38.2 (+1.1) | 60.5 (-11.6) | 71.0 (+16.0) | 65.0 (+0.7) | 43.4 (+13.8) | 32.1 (-3.3) | 88.2 (-0.2) | 86.7 (+6.1) |
| PointGroup† (2cm) Jiang et al. (2020) | 59.4 | 67.9 | 99.9 | 67.5 | 0.0 | 38.0 | 68.5 | 85.2 | 93.9 | 31.0 | 25.1 | 53.1 | 82.3 |
| GS³ + PointGroup (2cm) | 61.9 (+2.5) | 56.0 (-11.9) | 99.9 (+0.0) | 62.7 (-4.8) | 0.0 (+0.0) | 47.2 (+9.2) | 76.2 (+7.7) | 69.2 (-16.0) | 93.2 (-0.7) | 34.3 (+3.3) | 34.9 (+9.8) | 80.6 (+27.5) | 88.3 (+6.0) |

Table 5: Comparative 3D instance segmentation results for each category on the S3DIS Area-5 set. † denotes the reproduced results. The number in each bracket denotes the performance improvement (shown in red) or degradation (shown in blue) compared to the corresponding baseline.

Charles R Qi, Or Litany, Kaiming He, and Leonidas J Guibas. Deep hough voting for 3D object detection in point clouds. In *Proceedings of the IEEE/CVF International Conference on Computer Vision (ICCV)*, pp. 9277–9286, 2019.

Hugues Thomas, Charles R Qi, Jean-Emmanuel Deschaud, Beatriz Marcotegui, François Goulette, and Leonidas J Guibas. Kpconv: Flexible and deformable convolution for point clouds. In *Proceedings of the IEEE/CVF International Conference on Computer Vision (ICCV)*, pp. 6411–6420, 2019.

Siming Yan, Zhenpei Yang, Haoxiang Li, Chen Song, Li Guan, Hao Kang, Gang Hua, and Qixing Huang. Implicit autoencoder for point-cloud self-supervised representation learning. In *Proceedings of the IEEE/CVF International Conference on Computer Vision (ICCV)*, pp. 14530–14542, 2023.

Hengshuang Zhao, Li Jiang, Jiaya Jia, Philip HS Torr, and Vladlen Koltun. Point transformer. In *Proceedings of the IEEE/CVF International Conference on Computer Vision (ICCV)*, pp. 16259–16268, 2021.

Haoyi Zhu, Honghui Yang, Xiaoyang Wu, Di Huang, Sha Zhang, Xianglong He, Tong He, Hengshuang Zhao, Chunhua Shen, Yu Qiao, et al. Ponderv2: Pave the way for 3D foundataion model with a universal pre-training paradigm. *arXiv preprint arXiv:2310.08586*, 2023.

| Method | AP@50 | cabinet | bed | chair | sofa | table | door | window | bookshelf | picture | counter | desk | curtain | refrigerator | shower curtain | toilet | sink | bathtub | other furniture |
|---|---|---|---|---|---|---|---|---|---|---|---|---|---|---|---|---|---|---|---|
| PointGroup† (5cm) Jiang et al. (2020) | 49.1 | 48.5 | 70.3 | 77.0 | 64.9 | 66.2 | 38.3 | 24.8 | 45.3 | 15.1 | 25.5 | 30.2 | 27.5 | 54.9 | 54.2 | 94.8 | 39.8 | 76.9 | 29.7 |
| **GS³** + PointGroup (5cm) | 51.0 (+1.9) | 47.7 (-0.8) | 75.4 (+5.1) | 80.1 (+3.1) | 68.5 (+3.6) | 68.0 (+1.8) | 38.5 (+0.2) | 29.0 (+4.2) | 50.3 (+5.0) | 18.2 (+3.1) | 23.2 (-2.3) | 28.5 (-1.7) | 21.7 (-6.8) | 58.8 (+3.9) | 55.2 (+1.0) | 94.4 (-0.4) | 47.5 (+7.7) | 77.5 (+0.6) | 35.7 (+6.0) |
| PointGroup† (2cm) Jiang et al. (2020) | 57.6 | 49.9 | 72.5 | 87.1 | 59.6 | 67.2 | 48.5 | 38.7 | 61.2 | 32.0 | 21.8 | 28.5 | 43.6 | 54.4 | 70.0 | 98.3 | 69.4 | 79.4 | 54.7 |
| **GS³** + PointGroup (2cm) | 59.5 (+1.9) | 53.4 (+3.5) | 76.3 (+3.8) | 90.2 (+3.1) | 72.8 (+13.2) | 68.0 (+0.8) | 46.2 (-2.3) | 34.9 (-3.8) | 56.6 (-4.6) | 35.3 (+3.3) | 28.4 (+6.6) | 28.0 (-0.5) | 48.9 (+5.3) | 66.4 (+12.0) | 66.0 (-4.0) | 99.0 (+0.7) | 66.7 (-2.7) | 77.0 (-2.4) | 56.7 (+2.0) |

Table 6: Comparative 3D instance segmentation results for each category on the ScanNet v2 val set. † denotes the reproduced results. The number in each bracket denotes the performance improvement (shown in red) or degradation (shown in blue) compared to the corresponding baseline.