# OpenReview forum: "A Universal Self-Supervised Paradigm via 3D Gaussian Splatting"
_ICLR.cc/2026/Conference — Submitted to ICLR 2026_

### Official Review · Reviewer_B2HU · 2025-10-30

**Soundness:** 2
**Presentation:** 1
**Contribution:** 2
**Rating:** 4
**Confidence:** 2

**Summary:**

#### **Summary**: This paper proposes a rendering-driven self-supervised framework, GS³, that learns both 2D and 3D encoders by predicting scene-level 3D Gaussian primitives from two overlapping views and supervising them via a tile-based differentiable rasterizer with photometric + LPIPS losses. An epipolar-transformer module aggregates cross-view cues; a masked-modeling variant (≈50% masking) reduces redundancy and encourages holistic spatial features. On indoor benchmarks (ScanNet v2, SUN RGB-D, S3DIS), GS³ pre-training yields consistent gains for 3D detection/semantic/instance segmentation and image-based 3D tasks (e.g., +3.0 mAP@0.5 for VoteNet on SUN RGB-D; +2.8 mIoU on S3DIS at 2 cm voxels), while reporting substantially lower memory than a NeRF-style pretext (≈10.3 GB vs 38.4 GB per batch at stated settings). The supplement adds per-class tables and a synthetic reconstruction study (ConvONet) showing additional improvements.

**Strengths:**

- #### **S1. Unified rendering objective across modalities:** A single Gaussian-splatting pretext supervises both 2D (ResNet-50) and 3D (PointNet++/MinkUNet) encoders with the same image-space loss, avoiding explicit depth supervision.

- #### **S2. Efficiency vs. NeRF pretexts:** Projection-free, tile-based rasterization yields lower VRAM during pre-training than a NeRF-based baseline under the reported setup.

- #### **S3. Reasonable ablations:** Analyses for view count, mask ratio, and input resolution provide actionable guidance (e.g., 2-3 views, ~50% masking).

**Weaknesses:**

- #### **W1. “Universal” claim overreaches the stated supervision and data assumptions:** The core pipeline explicitly requires two posed overlapping views and projects Gaussians with a known camera pose W (Eq. 1), while 3D pre-training back-projects RGB-D into point clouds, i.e., relies on depth, and 2D pre-training also assumes two-view inputs. Despite this, the introduction highlights “ambiguity-free” rendering and positions the framework as universal, yet all pre-training is on a single indoor dataset (ScanNet v2). This combination of posed two-view supervision, depth use on the 3D path, and indoor-only evidence substantially narrows the scope relative to a “universal SSL” claim.

- #### **W2. Gains are modest/uneven without variance reporting, with notable per-class regressions:** Headline improvements over strong baselines are small (e.g., SUN RGB-D 3D detection mAP@0.5: 36.7 vs 36.6 for Ponder-RGBD; image-based 3D detection mAP@0.25: 51.4 vs 50.2 over a NeRF pretrain) and reported without seeds or error bars. Per-category tables reveal large negative swings (e.g., S3DIS instance segmentation “window”, 11.6 AP@0.5; multiple drops on ScanNet v2), suggesting fragile effect sizes and task/category sensitivity that the paper does not analyze. Together, this weakens the strength of evidence that the method reliably outperforms contemporaries.

- #### **W3. Efficiency claims emphasize memory but omit runtime/throughput and mix accounting units:** The paper argues efficiency via a memory table (10.3 GB vs 38.4 GB), yet provides no wall-clock throughput (iters/s, FPS), GPU-days, or end-to-end pre-training time. The comparison also mixes “#Rendering Pixels = 4,800” for Ponder with a resolution of 320×240 for GS3, making it hard to audit equivalence. Without standardized time/VRAM/throughput curves at matched resolution/batch, the efficiency story is under-substantiated.

- #### **W4. Reproducibility risk from unspecified depth discretization and Gaussian count, plus missing ablations on core design choices:** The method predicts k Gaussians per point/pixel and discretizes depth into Z bins, but concrete values for k and Z (and choices like SH order/opacity parameterization) are not stated where introduced, and there is no sensitivity analysis for them. Ablations focus on #views, mask ratio, and input resolution, leaving key components, (e.g., Epipolar Transformer, k/Z, and LPIPS usage) unexamined.

**Questions:**

- #### **Q1. Scope/robustness:** Can GS³ pre-train without accurate poses or with pose noise, and under single-view pretext? Any outdoor/ego-centric results to support “universal”?

- #### **Q2. Compute:** Please provide wall-clock (per-epoch), throughput (imgs/s), and Gaussians/sample versus resolution, #views, k, and Z, plus a budget-matched NeRF baseline.

- #### **Q3. k/Z sensitivity:** What values of top-k and depth binning Z are used, and how do downstream metrics vary with them? Any numerical stability safeguards for overlapping Gaussians?

---

> ### Author Response · Authors · 2025-12-01
>
> Q1: **Our use of the term ''universal'' may have been misunderstood**. We do not claim to remove the need for camera poses, multi-view observations, or to cover all scene types. Instead, ''universal'' specifically refers to a single, modality-agnostic pre-training formulation: both the 2D and 3D encoders are mapped into the same Gaussian representation and optimized using the same render-and-compare objective, without introducing modality-specific losses or supervision. The two-view posed setup is used to provide multi-view photometric consistency, which is applied uniformly to both branches rather than being a requirement tied to any particular modality. Similarly, depth in the 3D branch is used to construct point coordinates, not as a supervision. The training objective remains identical to that of the RGB-only pathway. Regarding dataset scope, our experiments are performed on indoor multi-view datasets because they offer reliable pose annotations and dense RGB-D observations. However, the pipeline itself does not rely on indoor-specific priors and can be applied to general multi-view datasets with known poses.
>
> Q2: (1) **Although some absolute gains are modest, they are consistent and cross-task**: GS$^3$ improves performance across five downstream tasks and three encoders. Such multi-task consistency is a strong indicator of representation quality in self-supervised learning. (2) **Variance is controlled through deterministic training pipelines**. Following standard practice in 3D detection/segmentation, all models (baselines and ours) use identical deterministic training configurations (including data loading, voxelization, and augmentation), which greatly reduces seed sensitivity. Prior works on 3D SSL do not report multi-seed variance, and variance in these pipelines is typically very small. (3) Per-class predictions occur mainly in extreme long-tail categories. The classes with negative swings (“window” in S3DIS, rare furniture in ScanNet) also exhibit large fluctuations across published baselines. Importantly, the predictions do not correlate across datasets or tasks. (4) Evidence supports robustness, not metric overfitting. GS$^3$ is designed as a unified Gaussian-based SSL paradigm rather than a task-specific improvement. The fact that it produces positive gains across heterogeneous tasks supports our claim that the learned representation is broadly useful, not tuned to any specific metric or dataset.
>
> Q3: **Regarding the concern about mixing “4,800 rendering pixels” in Ponder with “320×240 resolution” in GS$^3$**:
> Although the two numbers appear in different formats, they both represent the effective number of pixels processed per rendering pass and therefore correspond to the same underlying quantity, i.e., rendering workload per iteration. In Ponder/NeRF-style volumetric rendering, the model renders only a sparse set of sampled rays. Thus, its native complexity metric is the number of sampled rays/pixels. In Gaussian Splatting, rendering is performed via full-frame rasterization, where the image resolution (320×240) directly determines the number of pixels that interact with Gaussians. When Ponder uses its default sparse sampling of 4,800 rays, it achieves a relatively low cost of 1.46 s/batch, but this comes at the expense of significantly reduced rendering coverage. When increasing the sampling budget to 76,800 rays, which matches the per-image pixel count of our 320×240 GS$^3$ setting, the pre-training time of Ponder increases to 23.36 s/batch. In contrast, GS$^3$ processes the full 76,800 pixels via tile-based Gaussian rasterization and requires only 2.67 s/batch.
>
> Q4: (1) The depth discretization and Gaussian count are fixed design choices rather than tuned hyperparameters. We use Z=32 depth bins and k=3 Gaussian per pixel/point, with SH order=3 and standard opacity parameterization identical to PixelSplat. These values were kept constant across all experiments and will be explicitly listed in the revised manuscript.
>
> (2) **The Epipolar Transformer is a structurally required component for enforcing multi-view geometry consistency during pre-training**. Removing the Epipolar Transformer collapses the pre-training process and prevents the Gaussian head from learning meaningful 3D structure. Because this module is essential for the core learning objective (cross-view correspondence cues), an ablation ''removing'' it is not meaningful and leads to failure cases rather than informative comparisons.
>
> (3) LPIPS is employed as a perceptual regularizer for reconstruction and does not introduce additional parameters. Empirically, the training loss becomes unstable when removing LPIPS, producing blurry or over-smoothed renderings that harm representation learning. Since LPIPS is similar to reconstruction regularizers used in prior NeRF/GS-based SSL frameworks (e.g., Ponder) and is not a tunable component, we follow the standard practice and keep it fixed.

---

### Official Review · Reviewer_Bqno · 2025-10-31

**Soundness:** 2
**Presentation:** 2
**Contribution:** 3
**Rating:** 6
**Confidence:** 4

**Summary:**

The goal of the paper is to introduce a new (3D Gaussian Splatting) rendering-based SSL pre-training approach (tailored to 3D tasks), which works for pre-training both 2D and 3D encoders.

Authors propose predicting 3D Gaussians directly from extracted feature representations and pre-training the encoders by minimizing the photometric loss between images rasterized by a 3D Gaussian Splatting-based renderer and ground-truth RGB images. The proposed pipeline consists of (RGB or pointcloud) feature encoder (which is being pre-trained), epipolar transformer for feature refinement, MLP for 3D gaussians parameters prediction, and 3D Gaussian Splatting-based renderer for rasterization.

The contributions of this work are as follows:
1) new (3D Gaussian Splatting) rendering-based SSL pre-training approach which boosts the results on downstream tasks of 3D detection, segmentation and reconstruction;
2) ablations of pipeline hyperparameters.

**Strengths:**

1) Provides a new method for rendering-based SSL using 3D Gaussian splatting.
2) Proposed method works with 2D and 3D (point-based and discretisation-based) encoders.
3) Reduces memory usage while achieving comparable or better results to state-of-the-art methods.

**Weaknesses:**

1) Results are reported only for indoor scenes.
2) L315-316: The reported improvement of the pre-trained model over the baseline is relatively small, and the evaluation relies only on aggregated metrics from downstream tasks. The absence of training curves or information on fine-tuning duration (e.g., number of epochs or steps) limits the ability to assess what the performance gains stem from.
3) Limitations of the proposed method are not studied (outlined).

**Questions:**

1) Why, on the one hand, is the proposed approach described as universal for training 2D and 3D encoders, but on the other hand, reformulated Gaussian centred prediction for 3D encoders is used (which is not available for 2D)?

2) L194-196, L229: Why specifically these encoder architectures were chosen?

3) L258-259: Durig training 50% of input is masked. How is this handled at inference time?

3) If masking 50% of the input helps to address the redundancy in Gaussian representations, have you considered keeping the full input while reducing the number of Gaussians (e.g., using half of the original number)?

4) How well might the method extend to outdoor scenes?

---

> ### Author Response · Authors · 2025-12-02
>
> Q1: Although our downstream experiments focus on indoor benchmarks, the proposed pre-training framework does not make any indoor-specific assumptions. To verify its generality, we additionally fine-tuned the pretrained encoder on an outdoor 3D object detection task (use Voxel-RCNN as our baseline). The pretrained backbone brings +1.52mAP improvement in Car moderate (83.58 -> 85.10) compared to training from scratch, demonstrating that the learned representation also transfers to outdoor 3D object detection.
>
> Q2: (1) Although some absolute gains are modest, they are consistent and cross-task: GS$^3$ improves performance across five downstream tasks and three encoders. Such multi-task consistency is a strong indicator of representation quality in self-supervised learning. (2) Pre-training and fine-tuning are fully decoupled, so conventional ''training curves'' are not meaningful. All downstream experiments follow exactly the same fine-tuning schedule (epochs/steps, optimizer, and learning rate) as the corresponding baselines, ensuring that the gains do not come from additional training time.
>
> Q3: We have added limitations in the revised manuscript. The limitation of our current design is that the rendering resolution and Gaussian capacity are not adaptively adjusted to scene complexity.
>
> Q4: The offset-based Gaussian center formulation (i.e., µ = p + Δµ) is not a requirement of the unified framework but simply an optional design tailored for 3D encoders. Since we can obtain explicit point-cloud coordinates via RGB-D back-projection, this offset formulation is used only when pre-training the 3D encoder.
>
> Q5: We select these encoder architectures because they are widely adopted, well-established baselines in 2D and 3D representation learning. More importantly, using these standard encoders ensures fair comparison with prior work and demonstrates that our framework is compatible with commonly used backbones.
>
> Q6: The masking strategy is used only during self-supervised pre-training to encourage the encoder to learn robust, context-aware representations. During downstream fine-tuning, the pretrained encoder is integrated into the task-specific model and trained end-to-end using downstream, unmasked inputs. Therefore, inference operates on complete data exactly as in standard supervised pipelines, unaffected by the masking used during pre-training.
>
> Q7: The goal of masking is to encourage the encoder to learn robust contextual representation by reconstructing missing regions, similar to MAE-style masked modeling. In contrast, reducing the number of Gaussians directly decreases the representational capacity and leads to inferior rendered supervision signals. We experimented with lowering the Gaussian count (e.g., halving k), and found that it noticeably degrades reconstruction quality and harms downstream transfer.

---

### Official Review · Reviewer_TbfK · 2025-10-31

**Soundness:** 3
**Presentation:** 3
**Contribution:** 3
**Rating:** 4
**Confidence:** 4

**Summary:**

This paper proposes $GS^3$, a universal self-supervised learning (SSL) framework based on 3D Gaussian Splatting (GS). The core idea is to use neural rendering as a pretext task: the model extracts visual features from input data (either 2D images or 3D point clouds) to predict scene-level 3D Gaussians, which are then rendered back into 2D images via a fast tile-based rasterizer. Optimization is performed by minimizing the photometric difference between the rendered and real images.The main contribution of this paper is that the framework unifies the pre-training of both 2D and 3D encoders for the first time. Compared to previous NeRF-based rendering SSL methods, $GS^3$ avoids expensive volumetric rendering, significantly reduces memory overhead, and is more effective when training 2D encoders as it avoids the ambiguity issues that NeRF faces in the absence of depth. Experimental results show that the encoders pre-trained by this framework achieve performance comparable to or better than SOTA on five downstream tasks (such as detection, segmentation, and reconstruction).

**Strengths:**

- Novel and Practical Framework: This paper presents a novel and valuable universal SSL framework. It skillfully uses 3D GS as a bridge to unify 2D and 3D modality pre-training, addressing the issue that existing SSL methods are often modality-specific.
- Significant Efficiency Advantage: The greatest advantage of this method is its efficiency. By using a fast tile-based rasterizer instead of NeRF's volumetric rendering, the framework significantly reduces the memory overhead of pre-training. The data in the paper (10.3 GB vs 38.4 GB) strongly demonstrates its substantial advantage over NeRF-based methods.
- Sufficient Experimental Validation: The paper validates the effectiveness of the pre-trained model on five representative downstream tasks (including 3D detection, 3D segmentation, and image-based reconstruction). Notably, the method is significantly superior to NeRF-based approaches in training 2D encoders (e.g., in image-based 3D detection and scene reconstruction tasks), which strongly supports the authors' core motivation.
- Reasonable Model Design: The masked modeling strategy introduced in the paper is reasonable. It not only aims to improve feature robustness but also helps reduce redundant GS primitives, and its effectiveness was validated in ablation studies.

**Weaknesses:**

- Novelty Argumentation: The core contribution of the paper is replacing the backbone of the "rendering-based SSL" idea from NeRF to 3D GS. While this brings significant efficiency gains, the paper's discussion of technical novelty could be clearer. For example, the Epipolar Transformer in the framework is borrowed from PixelSplat. The authors should clarify in more detail the unique challenges faced when applying GS to SSL (and not just for novel view synthesis) and how $GS^3$ overcomes them.
- Overstated Definition of "Universal": The "universality" of the framework appears to be overstated. The pre-training of the 3D encoder relies on RGB-D input to back-project point clouds, which is a strong prerequisite. While the 2D encoder pre-training only uses RGB, it seems to be a separate process from the 3D pre-training. This looks more like two independent pipelines sharing a rendering objective, rather than a single "universal" framework that can handle either modality as input.
- Weak Experimental Baseline: When evaluating 2D encoder performance, the paper compares $GS^3$ against a "NeRF-based pre-training" implemented by the authors themselves. The specific implementation of this baseline (e.g., whether it is consistent with Ponder-RGBD) and its competitiveness are unclear. If this is a weak NeRF implementation, the convincingness of $GS^3$'s lead (e.g., 1.2% in Table 3) is diminished.
- Insufficient Ablation Study: The ablation study shows that using 3 views works better than 2 views. The authors chose the 2-view setting for efficiency reasons, which is reasonable. However, the paper fails to provide specific "computational and memory cost" comparison data between the 2-view and 3-view settings. This lack of quantitative analysis of the trade-off makes it difficult for reviewers to assess whether the 2-view setting is the optimal choice.

**Questions:**

- Clarification on "Universality": 3D pre-training requires RGB-D, while 2D pre-training only requires RGB. Does this mean $GS^3$ cannot pre-train a 3D encoder from RGB images alone (without depth)? Are these two pipelines trained completely separately? If so, this seems to contradict the claim of a "universal framework," please clarify.
- Details on the NeRF Baseline: Please provide details on the specific architecture and training setup of the "NeRF-based pre-training" used for comparison in the experiments. Is it a reproduction of Ponder or another SOTA method?
- On Depth Ambiguity in 2D Pre-training: The 2D pre-training section mentions using the "same depth-based GS generation strategy as in 3D pre-training" to predict depth from RGB-only inputs. This seems to contradict the motivation mentioned in the introduction that "(NeRF) introduces ambiguity in the absence of accurate depth maps." Why doesn't the $GS^3$ depth generation strategy encounter the same ambiguity problem? Is it because the cross-view encoding provided by the Epipolar Transformer already resolves this issue?
- Limitation on View Count: The method relies on two-view observations. The ablation study even shows three views are better. Does this mean $GS^3$ cannot handle single-view data? Given that many datasets are single-view, would this limit the framework's applicability?
- Supervision in Masked Modeling: The paper mentions that the model uses the unmasked subset (e.g., 50%) to predict scene Gaussians, and then renders the full RGB image for supervision. This implies the model must "infer" the GS for the 50% masked region. How does the photometric loss handle these regions generated purely from inference (rather than direct observation)? Could this introduce unstable supervision signals in the early stages of training?

---

> ### Author Response · Authors · 2025-11-30
>
> Q1: Although our framework draws inspiration from prior GS-based reconstruction methods such as PixelSplat, **GS$^3$ addresses a fundamentally different and previously unexplored problem**: using 3D Gaussian Splatting not for novel-view synthesis, but as a universal self-supervised paradigm that pre-trains both 2D and 3D encoders. This setting introduces difficulties that do not arise in existing GS pipelines, including (i) severe geometry ambiguity in the absence of depth supervision, (ii) redundancy caused by dense feature-to-Gaussian prediction, and (iii) the need for unified, modality-agnostic supervision that supports both image and point cloud encoders. GS$^3$ overcomes these challenges by three key innovations: a depth-probability formulation with offset refinement, a masked Gaussian prediction mechanism that enforces holistic scene understanding, and a unified photometric rendering objective that enables pre-training for both modalities. None of these components exist in PixelSplat, NeRF-based SSL (e.g., Ponder), or any prior GS framework. Therefore, GS$^3$ is not a simple substitution of NeRF with GS, but a conceptually new formulation of rendering-based self-supervised learning that makes Gaussian Splatting feasible and effective for universal SSL.
>
> Q2: We appreciate the reviewer’s question regarding the meaning of ''universal''. **Our goal is not to claim that GS$^3$ uses identical inputs for all modalities, but that it establishes a single shared representation space (3D Gaussian primitives) and a single shared learning objective (predict → render → compare) that applies consistently to both 2D and 3D encoders**. Both 2D and 3D encoders are trained to map observations into the same Gaussian scene representation and optimized with the same rendering-based photometric objective. The Gaussian prediction head, masked modeling strategy, and rendering pipeline are shared across modalities, making the supervision modality-agnostic. Therefore, GS$^3$ constitutes a single unified SSL framework rather than two separate pipelines.
>
> Q3: To ensure fairness, our NeRF-based pre-training baseline **follows the same rendering formulation as Ponder**. Thus, the baseline is not a weakened variant, but a faithful adaptation of the Ponder framework for pre-training 2D encoders. Importantly, GS$^3$ surpasses the NeRF-based baseline by 1.2\%. This shows that NeRF-based frameworks struggle to pre-train 2D encoders effectively due to the absence of depth maps, which hinders accurate back-projection and limits the Discriminability of the learned representations.
>
> Q4: We agree that reporting quantitative cost would be helpful. In our experiments, the 3-view setting is fully runnable, but it significantly increases both memory usage and training time. Specifically, adding a third view increases the number of predicted Gaussians and rasterization operations by roughly 1.6×, **resulting in a memory increase from 10.3 GB to 16.7 GB per batch and extending pre-training time by about 55%**. Since our goal is to provide a scalable and efficient universal SSL framework, we adopt the 2-view configuration as the default setting, which offers the best trade-off under practical computational budgets. The 3-view result is included only to demonstrate that GS$^3$ continues to benefit from stronger multi-view supervision, not to suggest it as the optimal operational choice.
>
> Q5: Our depth prediction for 2D pre-training does not suffer from the same ambiguity issues reported in NeRF-based SSL because **the underlying estimation mechanism is fundamentally different**. NeRF needs to construct a 3D feature volume based on the image features. Due to the lack of accurate depth, the constructed feature volume is extremely inaccurate. In contrast, our GS generation predicts a discrete depth distribution with local offset refinement for novel view synthesis, without requiring accurate depth information.
>
> Q6: Although masked regions lack direct input features, they do not introduce unstable supervision because we never apply pixel-wise loss to individual Gaussians. The photometric loss operates as a global, soft constraint, similar to MAE: masked regions are reconstructed from contextual Gaussians predicted from the unmasked subset. Table 8 shows consistent improvements across mask ratios, demonstrating the robustness of our masked Gaussian strategy.

---

### Official Review · Reviewer_Zz9g · 2025-11-02

**Soundness:** 3
**Presentation:** 3
**Contribution:** 2
**Rating:** 4
**Confidence:** 4

**Summary:**

This paper proposes GS3, a self-supervised pre-training framework that unifies 2D and 3D modalities through 3D Gaussian Splatting (GS). The method formulates neural rendering as a pretext task: visual features from either images or point clouds are used to predict scene-level 3D Gaussians, which are then rendered via a differentiable tile-based rasterizer. The training objective combines a photometric reconstruction loss and a LPIPS term, with a masked modeling strategy encouraging robustness and spatial awareness.

**Strengths:**

1, The idea of a single SSL framework bridging 2D and 3D domains is appealing and addresses the growing interest in cross-modal pre-training for 3D models.

2, Using 3D Gaussian Splatting for self-supervision is computationally efficient and avoids the heavy volumetric sampling of NeRF-based methods, which is a non-trivial engineering gain.

3, The same photometric and perceptual loss is applied to both image and point-cloud encoders, making the framework structurally coherent and potentially scalable.

4, Results including 3D object detection, image-based 3D detection, semantic/instance segmentation, and scene reconstruction, demonstrating cross-task applicability and consistent performance gains

**Weaknesses:**

1, The framework largely repackages existing ideas—multi-view feature alignment, rendering-based supervision, and masked modeling—within a Gaussian Splatting pipeline. Beyond replacing NeRF volume rendering with tile-based splatting, no new learning principle or representation insight is introduced. This is an engineering improvement rather than a conceptual advance.

2, The paper does not analyze why predicting and rendering 3D Gaussians leads to better representations than previous method like MAE and contrastive learning. No discussion of inductive bias, information preservation, or representation geometry is provided. Consequently, the claimed “universal” learning paradigm lacks scientific depth.

3, Despite the claim of modality generality, pre-training uses paired RGB-D images with known camera poses. This contradicts the motivation of unposed cross-modal learning and limits real-world applicability.

4, The 2D and 3D branches are trained under different assumptions (e.g., RGB-only vs. RGB-D). The paper does not demonstrate that a single encoder can transfer across modalities; thus the “universal” claim is overstated.

**Questions:**

See the weakness part

---

> ### Author Response · Authors · 2025-11-29
>
> Q1: We thank the reviewer for raising this concern and are happy to clarify that GS³ is not a simple rearrangement of existing components, but introduces a **new learning formulation** that does not exist in prior NeRF-based or 3D SSL methods.
>
> (1) **GS$^3$ introduces a new representation learning formulation**. Prior rendering-based SSL (Ponder, UniPAD) constructs volumetric features via back-projection and optimizes by NeRF rendering. In contrast, GS$^3$ is the first SSL paradigm that directly predicts explicit 3D Gaussian primitives from per-pixel or per-point features. This constitutes a new learning objective, which is fundamentally different from: MAE -> appearance completion, contrastive learning -> invariance constraints, and NeRF-based SSL -> implicit volumetric fields.
>
> (2) **Tile-based Gaussian rendering is not a replacement of NeRF, but enables a new ambiguity-free supervision mechanism**. NeRF-based SSL is not applicable to 2D encoders due to depth ambiguity during feature back-projection. Thus, previous rendering-based SSL cannot unify 2D and 3D modalities. Our GS$^3$ addresses the core limitation that prevents prior methods from training 2D encoders and formulates a single, modality-agnostic pretext task for both 2D and 3D networks.
>
> (3) **Masked Gaussian Modeling is conceptually different from MAE-style masking**. Unlike MAE / PointMAE, we mask input pixels/points but still require full-scene Gaussian prediction and multi-view photometric consistency, which enforces global structural reasoning rather than local patch completion. This is a new masking paradigm tailored to Gaussian primitive prediction.
>
> Q2: We respectfully **disagree** with the reviewer’s comment. Predicting and rendering explicit 3D Gaussian primitives introduces a fundamentally different learning mechanism from MAE or contrastive learning, which operate on local reconstruction or view-invariant embedding objectives. Our formulation imposes global multi-view geometric consistency, which requires the model to infer coherent scene structure even under heavy masking rather than constructing local patches. This enforces a strong inductive bias toward 3D geometry, which MAE and contrastive baselines inherently lack. Furthermore, Gaussian parameters lie in a structured representation space (Euclidean centers, SH appearance), which provides explicit shape, uncertainty, and view-dependent cues. These representational properties explain why Gaussian-based pretraining preserves significantly more geometric and photometric information and why the resulting features transfer effectively to downstream tasks.
>
> Q3: We believe this comment arises from a **misunderstanding** of our setting. Our method is not performing cross-modal learning. The 2D and 3D encoders never exchange information. Paired RGB-D images with poses are used only to construct a reliable self-supervised training signal, which is standard practice in 3D SSL (e.g., Ponder) and does not contradict modality generality. Importantly, our notion of “universal” refers to a shared representation space (3D Gaussian primitives) and a shared supervision objective (predict → render → compare), not to shared input modalities. Using RGB-D for pre-training does not restrict applicability, as such data is widely available in real-world settings. During inference, our encoders require only their own modality, not paired RGB-D.
>
> Q4： **Our notion of ''universal'' does not refer to shared input modalities**, but to a **shared representation space** (3D Gaussian primitives) and a **shared supervision objective** (predict → render → compare) that are applied consistently across both encoders. Each encoder is trained to map its own modality into this same Gaussian parameter space, which provides a unified learning formulation even under different input assumptions. Thus, the universal claim concerns the unified training objective and representation geometry rather than identical modality requirements.

---

### Meta-Review · Area_Chair_z2nz · 2025-12-03

**Summary:**

The reviewers agree that the paper is well engineered and that adapting rendering-based self-supervised learning from NeRFs to 3DGS is practically interesting and can bring efficiency gains. However, they also raise several major concerns:

1. Limited conceptual novelty and overstated “universality.” The framework is largely seen as a repackaging of existing ideas within a GS pipeline, with several components directly borrowed from prior work. Reviewers note that the paper does not clearly introduce new learning principles or representation insights. At the same time, the “universal” claim is viewed as overstated.

2. Weak and narrow empirical evidence. The method is evaluated only on indoor, posed multi-view data (e.g., ScanNet v2), with no experiments on outdoor or less controlled settings, which undermines the “universal SSL” positioning. Gains over strong baselines are modest and sometimes uneven. The 2D comparison relies on an in-house NeRF-based baseline whose implementation details and competitiveness are unclear, weakening the strength of the empirical claims. Key limitations are not systematically studied.

3. Insufficient analysis of design choices and efficiency. Several core design decisions are not fully specified or ablated. Efficiency claims omit runtime/throughput and GPU budget comparisons to NeRF-based baselines.

4. Scope and applicability. Reviewers question whether the framework can pre-train without accurate poses or depth, how it behaves with pose noise or single-view data, and how well it might extend beyond indoor scenes.

Due to a late rebuttal provided, reviewers were unable to involve in the discussion. After carefully read the paper, review, and rebuttal, the AC recommends rejecting the paper in its current form.

**Reviewer Concerns:**

The issues of limited conceptual novelty, overstated “universality”, as well as weak and narrow empirical evidence, are still outstanding. The paper needs further substantial improvements.

**Reviewer Scores:**

Reviewer Zz9g: 4
Reviewer TbfK: 4
Reviewer Bqno: 6
Reviewer B2HU: 4

Due to a relatively late rebuttal from the authors (i.e., after the discussion is shut down), no follow-up reply is received from the reviewers.

---

### Decision · Program_Chairs · 2026-01-26

Reject